# Evaluation of Performance of Inexpensive Laser Based PM$_{2.5}$ Sensor Monitors for Typical Indoor and Outdoor Hotspots of South Korea

**Sungroul Kim *** **, Sujung Park and Jeongeun Lee**

Department of Environmental Health Sciences, Soonchunhyang University, Asan 31538, Korea;
psj57732398@gmail.com (S.P.); l01025263029@gmail.com (J.L.)
\* Correspondence: Sungroul.kim@gmail.com; Tel.: +82-41-530-1266

**Featured Application: In consideration of relatively stable outcomes with the application of a correction factor for relative humidity, recently introduced inexpensive real-time monitors (IRMs), ESCORTAIR (ESCORT, Seoul, Korea) or PurpleAir (PA) (PurpleAir U.S.A.), our study supports their usage in PM$_{2.5}$ monitoring for various urban hotspots.**

**Abstract:** Inexpensive (<\$300) real-time particulate matter monitors (IRMs), using laser as a light source, have been introduced for use with a Wi-Fi function enabling networking with a smartphone. However, the information of measurement error of these inexpensive but convenient IRMs are still limited. Using ESCORTAIR (ESCORT, Seoul, Korea) and PurpleAir (PA) (PurpleAir U.S.A.), we evaluated the performance of these two devices compared with the U.S. Environmental Protection Agency (EPA) Federal Equivalent Monitoring (FEM) devices, that is, GRIMM180 (GRIMM Aerosol, Germany) for the indoor measurement of pork panfrying or secondhand tobacco smoking (SHS) and Beta-ray attenuation monitor (BAM) (MetOne, Grants Pass, OR) for outdoor measurement at the national particulate matter (PM$_{2.5}$) monitoring site near an urban traffic hotspot in Daejeon, South Korea, respectively. The PM$_{2.5}$ concentrations measured by ESCORTAIR and PA were strongly correlated to FEM (r = 0.97 and 0.97 from indoor pan frying; 0.92 and 0.86 from indoor SHS; 0.85 and 0.88 from outdoor urban traffic hotspot). The two IRMs showed that PM$_{2.5}$ mass concentrations were increased with increased outdoor relative humidity (RH) levels. However, after applying correction factors for RH, the Median (Interquartile range) of difference compared to FEM was (14.5 (6.1~23.5) %) for PA and 16.3 (8.5–28.0) % for ESCORTAIR, supporting their usage in the home or near urban hotspots.

**Keywords:** PM$_{2.5}$; sensor; correction; pan frying; secondhand smoke; urban traffic

## 1. Introduction

A large volume of previous epidemiological studies relied on the use of ground-based fixed national monitoring stations [1,2]. However, recently, inexpensive (<\$300) particulate matter (PM) monitors (PM) have been introduced for home usage in South Korea. These devices can provide PM distribution patterns at high temporal and spatial resolution [3–5] which is a substantial improvement on establishing a pollution monitoring networking system as well as environmental epidemiologic study [6], as compared to traditional approaches that relied on relatively small number of ground-based fixed national air monitoring stations or mobile sampling techniques.

Most of these low-cost devices are classified into two groups, that is, optical particle counters (OPCs) or photometers. OPCs use the light scattered from individual particle to estimate the concentration

of particles in different size ranges [7,8]. These data, along with assumptions of the particle shape and density, can be converted to estimate mass concentrations that compare favorably with reference instruments [7,8]. However, it has been reported that there may be bias when aerosol type or size is unknown [7,8].

Photometers use a light source to illuminate sensing zones that contain many particles at one time [9,10] and obtained that the mass concentration of aerosol scales linearly with the amount of light scattered by an assembly of particles captured at a discrete angle from the incident light [11]. The light scattered by the assembly of particles is measured by a photodetector at an angle specific to the photometer model, often 90° from the incident light [12]. The intensity of scattered light is directly proportional to gravimetrically measured mass concentration, although the relationship is dependent on the light scattering characteristics, density and size distribution of the particle [12].

The cost of research grade light scattering instruments (approximately, $10,000 or higher) limits their use to conduct studies at high temporal and/or spatial resolution. In Korea, a laser-based inexpensive OPC based IRM, for example, ESCORTAIR, was recently introduced. However, the reliability of ESCORTAIR have been unknown. Therefore, it may be necessary to test this new device before it is applied in a high spatial-temporal resolution exposure assessment study. As a proper calibration protocol providing a correction factor can have a dramatic impact on precision, accuracy and bias of a real-time monitor, researchers evaluated the OPC or photometers for use in the laboratory, outdoors or in the home in other countries [3,9,13–16]. A recent article reported that "one size fits all" approach to obtain $PM_{2.5}$ mass concentrations by OPC result in relatively high uncertainty in complex exposure situations. Although OPC Therefore, corresponding conversion curve approach may be most valuable when a relatively high contrast is expected in exposure levels for example, daytime home with indoor combustion sources, BBQ or secondhand smoke versus night time or day time outside with heavy traffic volume versus night time [16]. To our knowledge, no one has rigorously evaluated the performance of IRMs, operated with OPC or photometer, in Korea with comparison of the U.S. Federal Equivalent Method (US FEM) [17].

In this study, we evaluated the performance of inexpensive (less than $300) real-time PM monitors (IRMs), with high cost (about $2000–$10,000) and cross-comparisons between them and research grade PM monitors (RGMs). We used US FEMs as reference instruments (approximately $20,000 or higher) and provided a final error of mass concentration ($PM_{2.5}$) measurement after applying correction factors in this study.

## 2. Materials and Methods

### 2.1. $PM_{2.5}$ Real-Time Monitors

A laser-based light-scattering $PM_{2.5}$ sensor monitor (ESCORTAIR, ESCORT, Seoul, Korea) (weight <300 g, volume <510 cm$^3$) consisted of an optical particle counting (OPC) PM sensor (INNOSIPLE1), $CO_2$ sensor, temperature relative humidity sensor, data transfer networking module and light-emitting diode (LED) display screen (Figure 1). In the ESCORTAIR, the sensing volume is illuminated with a laser and airborne particles are counted and processed one at a time. There were various IRMs commercially available in South Korea. In this study, however, we chose ESCORTAIR as they allowed us to directly transfer data to our data server using its Wi-Fi function.

For comparison purposes, another inexpensive photometer type of PM monitor (PA, PurpleAir, Draper, UT, USA) (https://www.purpleair.com/), mounting two Plantower sensors in a monitor, was used. PA is recommended its usage by AQ-SPEC (Air quality sensor performance evaluation center, South Coast Air Quality Management District, CA, USA) or US EPA (Environmental Protection Agency, NC, USA) as an IRM. The inlet system of these IRMs did not have an impactor or a cyclone unlike that of the RGM.

The performance of the two IRMs (one OPC, that is, ESCORTAIR and one photometer, that is, PA) costed less than $300 were simultaneously compared with those of high-cost devices ($10,000 or so),

that is, research-grade laser photometers including PDR-1500 (Thermo Scientific, Waltham, MA, USA) and SIDEPAK AM510 (TSI, Inc., Shoreview, MN, USA).

These research-grade monitors have a cyclone inlet or an impact inlet for the measurement of the respirable fraction of airborne particulate matters in different environments and can provide real-time data. The SIDEPAK is a portable battery-operated personal aerosol monitor with an impact inlet and light-scattering laser photometer that provides real-time aerosol mass concentration. The PDR-1500 (Personal DataRAM 1500) is a nephelometric monitor with a cyclone inlet for the measurement of the respirable fraction of the airborne particulate matters. The PDR-1500 can simultaneously collect particles on a 37 mm filter for the gravimetric analysis by passing through the sensing zone.

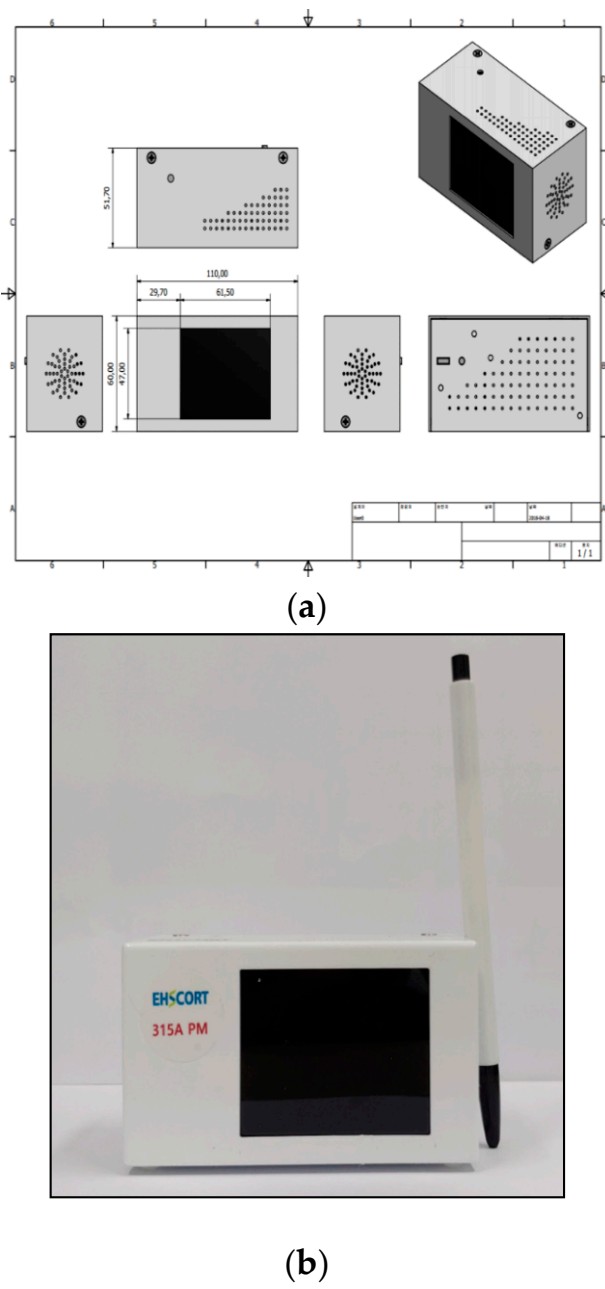

(a)

(b)

**Figure 1.** Layout of ESCORTAIR installed INNOSOPLE-1 PM sensor for measuring particulate matter (PM) concentrations.

## 2.2. Federal Equivalent Method

As mentioned above, to conduct a comparative measurement of IRMs, we used the U.S. Environmental Protection Agency (EPA) Federal Equivalent Method (FEM), that is, GRIMM 180 (GRIMM Aerosol, Technik Ainring GmbH & Co. KG, Ainring, Germany) for indoor testing and BAM-1022 (MetOne, Grants Pass, OR, USA) for outdoor field tests [17].

The Grimm Technologies, Inc. Model EDM 180 PM2.5 Monitor is a light scattering OPC monitor operated for 24 h at a volumetric flow rate of 1.2 L/min, configured with a Nafion®- type air sample dryer. BAM-1022, a beta-ray attenuation mass monitor has a $PM_{2.5}$ particle size separator. Using BAM, we obtained 24 1-h average measurements at the national $PM_{2.5}$ monitoring supersite operated by National Institute of Environmental Research, Daejeon, Korea.

## 2.3. Flow Rate Inspection

Before each indoor and outdoor experiment, RGMs, that is, the PDR-1500 and SIDEPAK were zeroed with an in-line high efficiency particulate air (HEPA) filter and the flow of each device (1.52 L/min, 1.7 L/min and 0.5 L/min) was checked by a mass flowmeter (TSI, Inc., Shoreview, MN, USA). The temperature and relative humidity data were also downloaded from the devices if the device is equipped with sensors for these data. Flow rate of GRIMM, one of FEMs, was also checked with a similar way. The values from BAM, the other FEM, were used as reference, since it is located at the KOREA national $PM_{2.5}$ monitoring site (Daejeon) and operated with high QAQC programs [18,19] to report hourly outdoor $PM_{2.5}$ data.

## 2.4. Experimental Setting

We collected $PM_{2.5}$ concentrations by performing both an indoor exposure test on March 2018 and outdoor $PM_{2.5}$ monitoring at the national supersite located in Daejeon, Korea from June to July 2018 including rainy days. We used the 2 sets of each IRM or RGM for indoor and outdoor testing (serial numbers of devices: Table 1).

### 2.4.1. Indoor Test

The indoor test included scenarios of frying pork in a pan and exposure to secondhand smoke (SHS). Indoor pan-frying tests were conducted at inside of an empty laboratory (4 m × 10 m × 3.5 m, W D H), according to the protocol described in detail in our previous article [20]. In brief, this experiment was carried out over a 2 h measurement period per trial including first 9 min simulating the barbequing of pork belly (100 g). Standard operating protocol: a portion of pork belly (100 g) was pan-fried for 9 min: 3 min on Side A, 3 min on Side B; then 1.5 min on Side A again and finally 1.5 min on Side B again. When we fried pork belly, after the first 9 min, we opened a window (0.5 m × 0.8 m) to allow the ventilating air to naturally reduce the $PM_{2.5}$ concentration.

Measurement of the $PM_{2.5}$ levels from the exposure to secondhand smoke was conducted with a lighted cigarette burned. We opened the same window after 30 min during our secondhand smoke exposure level test. Then, we collected concentration data over next two hours.

During our indoor test, we maintained a minimum distance of 20 cm among these devices, at least 50 cm from the emission sources and 1 m above the floor. To minimize the effect of additional source contribution to our $PM_{2.5}$ measurement results, we reported our $PM_{2.5}$ results after subtracting the field background $PM_{2.5}$ concentrations measured at the baseline. We collected with the frying pan test and the secondhand exposure test three times with GRIMM 180 on separate days. The indoor test data from each device (80-s interval for PA, 60 s for the remaining devices) were calculated to the 5 min average level to be compared with the outcomes from GRIMM. The final number of data for the 2 h panfrying test was approximately 50 (2 sets of each device × 12 data point/h × 2.0 h) and that for the 2.5 h secondhand smoke exposure test was about 60 (2 sets of each device × 12 data point/h × 2.5 h) for $PM_{2.5}$, as well as temperature and relative humidity.

**Table 1.** Summary of PM$_{2.5}$ measurement range of concentration, measurement interval, weight and Wi-Fi availability reported by manufacturers and unit cost in South Korea by 30 June 2018.

| | Device Classification [a] | Sensor Type [b] | Measurement Range | Sampling Pump Flow Rate | Precision [c] | Log Interval [c] | Unit Price ($) | Weight (g) | Wi-Fi |
|---|---|---|---|---|---|---|---|---|---|
| GRIMM (EDM180) [1] (GRIMM Aerosol, Germany) S/N #: 11R15047 | FEM | OPC | 0~3,000,000 particles/Liter | 1.2 L/min, | 97% over the whole measuring range | 5 s to 1 h | 19,000 | 20,000 | No |
| BAM-1020 [2]. (MetOne, OR) S/N #: N11181 | FEM | Beta ray Attenuation | 0~1000 mg/m$^3$ | 16.7 L/min | Exceeds US-EPA Class III PM$_{2.5}$ FEM standards | 1 min to 1 h | 23,750 | 24,500 | No |
| ESCORTAIR [3] (ESCORT, Seoul, South Korea) S/N #: 6a:c6:3a:c7:83:bf 6a:c6:3a:c7:88:b1 | IRM | OPC | 1000 µg/m$^3$ | NA | ±10%@100~500 µg/m$^3$ | 30 s | 300 | 400 | Yes |
| PA [4] (PurpleAir, CA, USA) S/N #: A0:20:A6:A:AD:1B. A0:20:A6:B:83:32 | IRM | Photometer | 0~500 µg/m$^3$ as effective range | NA | ±10%@100~500 µg/m$^3$ | 80 s | 300 | 450 | Yes |
| PDR-1500 [5] (Thermo Scientific, MA, USA) S/N #: CM17422007, CM17422017 | RGM | Photometer | 0.001~400 mg/m$^3$ | Adjustable 0 to 3.5 L/min | ±2% of reading or ±0.005 mg/m$^3$ | 1 s to 1 h | 9000 | 1200 | No |
| SIDEPAK [6] (TSI, MN, USA) S/N #: 11104037, 11008055 | RGM | Photometer | 0.001~100 mg/m$^3$ | Adjustable 0 to 1.8 L/min | ±0.001 mg/m$^3$ over 24 h as zero stability | 1 s to 60 s | 6000 | 460 | No |

[a] Federal equivalent method (FEM), Research Grade Monitor (RGM) and inexpensive real-time monitor (IRM); [b] Optical particle count (OPC); [c] Information from manufacture. 1. GRIMM: https://www.grimm-aerosol.com/fileadmin/files/grimm-aerosol/General_Downloads/The_Catalog_2018_web.pdf. 2. BAM: https://metone.com/wp-content/uploads/2017/08/bam-1020-9803_touch_screen_manual_rev_k.pdf. 3. ESCORTAIR: This study. 4. PA: https://www.purpleair.com/sensors. 5. PDR: https://www.newstarenvironmental.com/air-toxic-monitors/personal-dataram-pdr1500-aerosol-monitor.html?_vsrefdom=adwords&gclid=CjwKCAiA2fjjBRAjEiwAuewS_cYiCFBPpx0d0bTaRKtmxe-1Kt22JVs352pQKq9e63XyqT_pIbAZsRoCn9UQAvD_BwE. 6. SIDEPAK https://www.tsi.com/getmedia/84b5be22-c339-49bc-ab97-e3c4baee16c1/SidePak%20AM520_US_5001737_Web_1.

### 2.4.2. Outdoor Test

Using the same configuration of monitoring devices, we also measured the ambient $PM_{2.5}$ concentration at the outdoor Roof-top of one of the national $PM_{2.5}$ Supersites located in Daejeon, South Korea, operating BAM (MetOne, Grants Pass, OR, USA). Main body of BAM was installed at inside of an experiment laboratory of the Supersite while the inlet of BAM was located at the Roof-top. The outdoor temperature and RH during were measured by the sensors in ESCORTAIR, PA and PDR-1500 and the values were crosschecked. Measurements from IRMs were collected every minute, except for PA, which provided a response every 80 s. To compare the hourly concentration values provided by the Supersite, we calculated the hourly mean values using measured values acquired at hourly intervals from each IRM device. Final sample size for the outdoor data was 240 (2 sets of each device × 24 data points/day × 5 days) for $PM_{2.5}$, as well as temperature and relative humidity.

### 2.5. Statistical Analyses

The Spearman correlation tests were used to evaluate the associations among the outcomes of devices measured at indoor or outdoor environments, considering that variables were not normally distributed.

Using outdoor measurement data, we evaluate the association of device response with various relative humidity level. We also evaluated the associations of the daily mean concentrations from IRMs with those obtained with FEM methods using multivariate linear regression models. We used the values obtained with the FEM as dependent variables and those obtained with real-time devices as independent variables to obtain a correction factor. The hourly mean temperature and relative humidity data for each sampling date, which was previously compared with the nearest national meteorological monitoring sites, were used to adjust the effects of relative humidity on the association between IRMs and FEM outcomes. The corresponding slopes and coefficients of determinant (R2) for each RT monitor were also provided. The final measurement error (%) was calculated based on the U.S. EPA performance evaluation program for PM instruments;

$$Difference_i = \frac{Measurement_i - FEM_i}{FEM_i} \tag{1}$$

EPA specifies that the percent bias goal for acceptable measurement uncertainty should be within ±20% [17]. Here, we reported median (IQR) value of the differences per device because the distribution of differences was not normal. We also provided mean of the differences just to compare with value in the guideline [21]. All analyses were conducted with SAS (Version 9.4) and R software (Version 2.15.3, R Development Core Team).

## 3. Results

### 3.1. $PM_{2.5}$ Concentration

In our indoor test, the median (IQR) $PM_{2.5}$ concentration over the pan-frying test was 86.8 μg/m³ (17.8–254.4 μg/m³) by the real-time ESCORTAIR, 104.9 μg/m³ (43.9–228.2 μg/m³) and 236.2 μg/m³ (49.3–648.7 μg/m³) by the real-time PA or PDR-1500 devices and 153.2 μg/m³ (46.2–409.7 μg/m³) by using GRIMM (Table 2). The median (IQR) concentrations for secondhand smoke (SHS) test were 20.9 (17.4– 156.6) μg/m³, 31.2 (14.4–194.3) μg/m³ and 28.4 (12.8–314.0) μg/m³ for ESCORTAIR, PA and PDR, respectively, whereas GRIMM provided 23.5 (15.9–107.1) μg/m³.

Simultaneous outdoor $PM_{2.5}$ monitoring results are also provided in Table 2. The median (IQR) of the hourly average values of ESCORTAIR and PA were 13.7 (7.3~21.2) and 19.7 (9.3~35.8) μg/m³, which were an overestimation of the values obtained by BAM, one of U.S. EPA FEMs. During indoor testing, the median temperature and RH were approximately 20~22 °C and 37%. During outdoor testing, the median values (IQR) were 30.7 (25.6~40.7) °C and 56.4 (34.8~71.4) %, respectively (Table 2).

**Table 2.** PM$_{2.5}$ mass concentration (µg/m$^3$) (median, IQR) measured by real-time sensor devices and the federal equivalent method as well as the temperature and relative humidity throughout the sampling period.

| | Indoor—Pan-Frying (n = 50) | Indoor—SHS (n = 60) | Outdoor—Urban Traffic Hotspot (n = 240) |
|---|---|---|---|
| GRIMM | 153.2 (46.2–409.7) | 23.5 (15.9–107.1) | NA |
| BAM | NA | NA | 9.0 (4.0–22.0) |
| ESCORTAIR | 86.8 (17.8–254.4) | 20.9 (17.4–156.6) | 13.7 (7.3–21.2) |
| PA | 104.9 (43.9–228.2) | 31.2 (14.4–194.3) | 19.7 (9.3–35.8) |
| PDR-1500 | 236.2 (49.3–648.7) | 28.4 (12.8–314.0) | 13.8 (6.8–34.8) |
| SIDEPAK | 261.3 (71.5–800.0 | 50.0 (21.0–652.0) | 29.1 (15.6–59.9) |
| Temp. (°C) | 21.7 (21.1–21.7) | 20.1 (19.7–20.5) | 30.7 (25.6–40.7) |
| RH (%) | 37.0 (35.0–39.0) | 37.0 (35.0–38.0) | 56.4 (34.8–71.4) |

## 3.2. Correlations among Devices and the Fem

High-level correlations were obtained between the IRMs (ESCORT, PA) and FEM. The Spearman correlation coefficients were 0.97 (P = 0.0001) or 0.92 (P = 0.0001) between ESCORTAIR and GRIMM for indoor pan-frying or the SHS test, respectively. The correlation coefficients between PA and GRIMM were similar (0.97 (P = 0.0001) or 0.86 (P = 0.0001)) for the indoor test (Table 3). Our outdoor test also showed high correlation between ESCORTAIR and BAM (0.84 (P = 0.0001)). The correlation coefficient for PA was 0.88 (P = 0.0001).

Similarly, the measurements by RGMs (PDR-1500, SIDEPAK) and FEM were highly correlated: 0.97–0.98 for the Pan-frying test, 0.88~0.96 for the SHS test and 0.84–0.91 for the outdoor test. Between IRMs, that is, ESCORTAIR and PA, the association of measurements were strong to each other (r = 0.93: Indoor pan frying; 0.85: Indoor-SHS; 0.93: Outdoor-urban traffic). In addition, they showed similar correlation patterns to PDR-1500 or SIDEPAK (Table 3).

**Table 3.** Scatter plots and Spearman correlation coefficients between real-time PM$_{2.5}$ monitoring devices and FEM.

| | Indoor-Pan-Frying | | | | | Indoor-SHS | | | | | Outdoor Urban Traffic Hotspot | | | | |
|---|---|---|---|---|---|---|---|---|---|---|---|---|---|---|---|
| | FEM | E | PA | P | S | FEM | E | PA | P | S | FEM | E | PA | P | S |
| FEM | 1 | | | | | 1 | | | | | 1 | | | | |
| E | 0.97 | 1 | | | | 0.92 | 1 | | | | 0.85 | 1 | | | |
| PA | 0.97 | 0.93 | 1 | | | 0.86 | 0.85 | 1 | | | 0.88 | 0.93 | 1 | | |
| PDR | 0.98 | 0.95 | 0.99 | 1 | | 0.96 | 0.93 | 0.94 | 1 | | 0.84 | 0.93 | 0.99 | 1 | |
| S | 0.98 | 0.99 | 0.96 | 0.98 | 1 | 0.88 | 0.86 | 0.88 | 0.93 | 1 | 0.91 | 0.91 | 0.99 | 0.99 | 1 |

FEM: Federal Equivalent Method: In this study Indoor: GRIMM Optical particle courting, OPC), Outdoor: BAM (Beta ray attenuation monitor). E: ESCORTAIR, PA: PurpleAir, PDR: PDR-1500, S: SIDEPAK.

## 3.3. Effects of Ambient Humidity for Outdoor Measurement

In this study, as seen in Supplementary Figure S1, we observed that the PM$_{2.5}$ concentration of IRMs were significantly increased with the increase in relative humidity level. The slope obtained from a simple regression line of FEM on IMRs, that is, (BAM = Slope * ESCORTAIR + intercept) at a relative humidity above 80%, was smaller than the slope at < 20.0%, 20.1~40.0%, 40.1~60.0%, 60.1~80.0%. The degree of decreasing trend was larger with IRMs, compared to two RGMs (Figure 2).

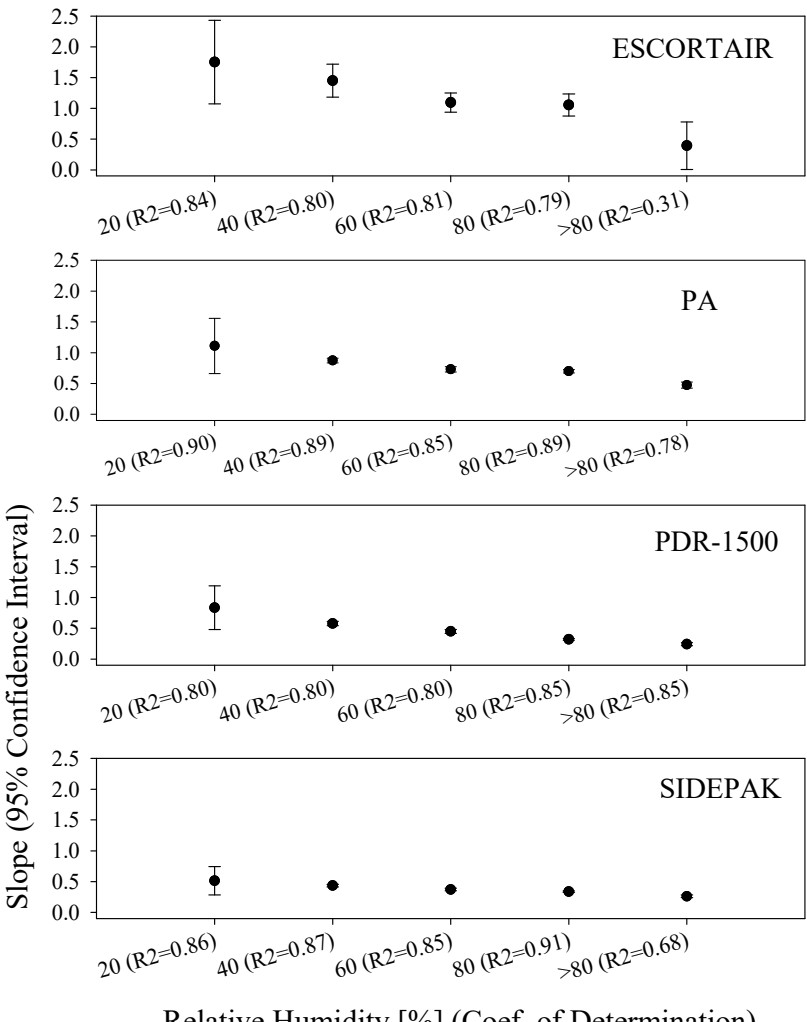

**Figure 2.** Change of the slope with 95% confidence interval obtained from the regression line of BAM (Beta ray attenuation monitor) on IRMs (Inexpensive real-time particulate matter monitors) or RGMs (research grade monitors) by the degree of relative humidity.

*3.4. Correction Factor*

Since we found the effect of RH on the measurements of ESCORTAIR or PA, we developed our own correction models for those IRM devices by performing stepwise linear regression. The correction factors were obtained from the regression models (1.11 for ESCORTAIR and 1.92 for PA, $p < 0.01$), (1.00 for ESCORTAIR and 0.87 for PA, $p < 0.01$) and (1.15 for ESCORTAIR and 0.70 for PA, $p < 0.01$) for the measurement of $PM_{2.5}$ resulting from indoor pan-frying, SHS or the urban traffic hotspot of South Korea, respectively.

After adjusting for temperature and relative humidity, the results were unchanged for the indoor tests (1.10 for ESCORTAIR and 1.90 for PA, $p < 0.01$), (0.97 for ESCORTAIR and 0.81 for PA, $p < 0.01$) (Table 3). For the outdoor measurement, the change of slopes (0.72 for ESCORTAIR and 0.77 for PA, $p < 0.01$) was relatively small but the R2 values were changed (Table 4).

**Table 4.** Slopes obtained from stepwise linear calibration models with adjusted $R^2$ (Dependent variable: US EPA FEM, Independent variable: IRM).

| | Indoor—Pan-Frying | | | | Indoor—SHS | | | | Outdoor—Urban Traffic Hotspot | | | |
|---|---|---|---|---|---|---|---|---|---|---|---|---|
| | Single | | Multivariate * | | Single | | Multivariate * | | Single | | Multivariate * | |
| | β | $R^2$ | β | $R^2$ | β | $R^2$ | β | $R^2$ | β | $R^2$ | β | $R^2$ |
| ESCORTAIR | 1.11 | 0.98 | 1.10 | 0.98 | 1.00 | 0.92 | 0.97 | 0.92 | 1.15 | 0.70 | 1.14 | 0.81 |
| PA | 1.92 | 0.94 | 1.90 | 0.94 | 0.87 | 0.89 | 0.81 | 0.90 | 0.70 | 0.83 | 0.71 | 0.87 |
| PDR-1500 | 0.33 | 0.98 | 0.33 | 0.98 | 0.54 | 0.91 | 0.49 | 0.92 | 0.33 | 0.72 | 0.36 | 0.80 |
| SIDEPAK 1 | 0.34 | 0.98 | 0.32 | 0.99 | 0.28 | 0.90 | 0.31 | 0.92 | 0.35 | 0.84 | 0.36 | 0.89 |

\* results obtained after adjusting for temperature and relative humidity.

### 3.5. Bias after Application of Correction Factors

We then determined the extent to which the original $PM_{2.5}$ data measured by ESCORTAIR were improved after applying the correction factor obtained from the model by performing comparative analyses using the outcomes of multivariate regression models (Table 4). Using the corrected data, the final coefficient of determination (R2) between FEM (y) and ESCORTAIR (x) was 0.81. The coefficient for PA was 0.87. We found the difference (median (IQR)) with the calibrated data, compared to FEM, to be 16.3. (8.5~28.0)% for ESCORTAIR and 14.5 (6.1 to 23.5)% for PA for outdoor environments (Figure 3). The bias (mean of the difference) was 13.1% for ESCORTAIR and 7.8% for PA for outdoor. The bias for indoor data was at least similar or lower than the bias level obtained from the outdoor test.

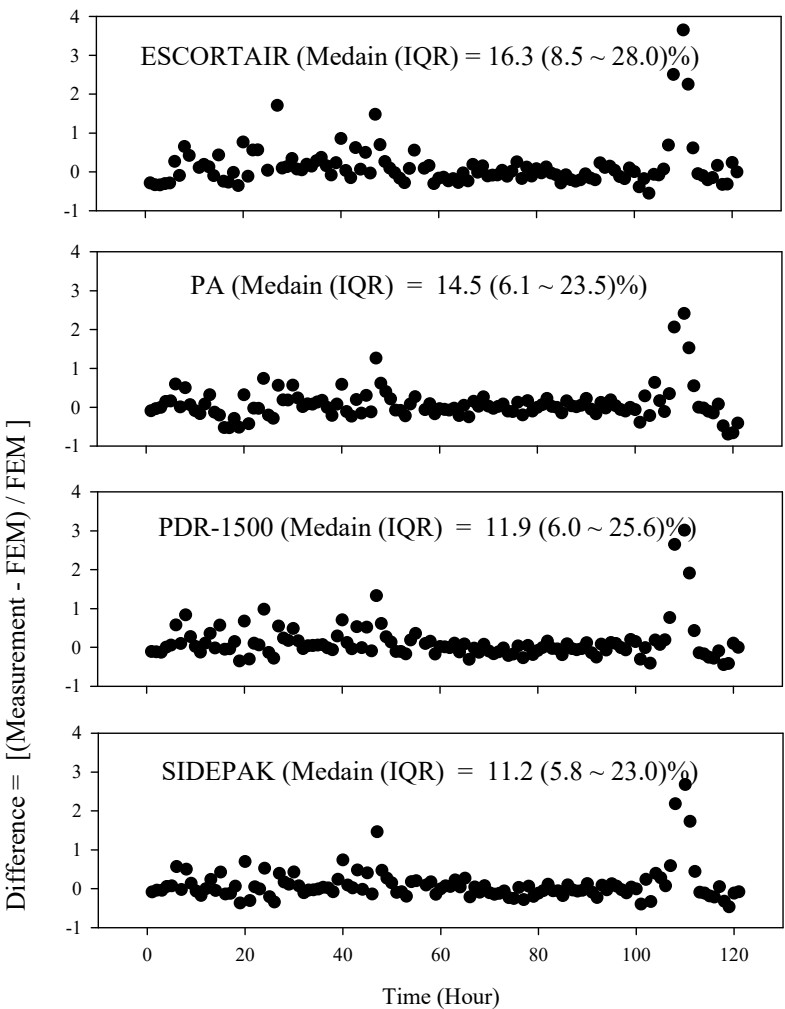

**Figure 3.** Distribution of difference of outdoor measurements against the FEM; BAM for outdoors.

## 4. Discussion

In this study, we compared PM$_{2.5}$ concentrations measured with IRMs and RGMs to those measured with U.S. EPA FEMs. A relationship between the real-time PM$_{2.5}$ concentration and FEM were acceptable (R = 0.97 to 0.98: Indoor pan frying; 0.86 to 0.96: Indoor-SHS; 0.84 to 0.91: Outdoor-urban traffic, respectively).

This study conducted indoor testing with common PM$_{2.5}$ sources of Korea, that is, frying pork in a pan or smoking. The indoor test was conducted in an indoor laboratory in which the room temperature was maintained at 20~22 °C and the relative humidity at 37% because we assumed that most low-cost PM sensing devices would be established inside susceptible populations' homes where the indoor temperature and humidity level would be relatively stable, compared to the outdoor environment.

In addition, in this study, we extended our comparison test by performing them outside with a federal equivalent method. We discovered that, on days with a high level of outdoor RH (80% or higher), our IRMs overestimated the PM$_{2.5}$ level. Thus, finally, we got the correction model providing a correction factor for ESCORTAIR to adjust for the effect of temperature and relative humidity as such a process has been conducted in a regulatory monitoring site (South Coast Air Quality Monitoring District, SCAQMD) for the field use of these kinds of inexpensive sensing devices including PA [22,23].

Several previous studies provided a correction factor for SIDEPAK monitors: 0.77 in Northern California, U.S.A. (ambient air), 0.43 or 0.52 in Italy (ambient air in urban or rural areas) and 0.42 in Italy (indoor-outdoor mixed environment) [24–26], which were comparable to our results (0.36). Our correction factor (0.36) for PDR-1500 was smaller than results reported by Wang et al. (2016) [27] (0.71 compared to PDR-1500 using its own filtering method) but very similar to the values obtained by Ramachandran et al. (2000) (0.33) and Wallace et al. (2011) (0.38), who conducted their studies on atmospheric environments [28,29].

A lack of quantitative information on the speciation of particles, traffic volume, type of vehicle or the particular sampling time or season limits further exploration of the basis for the differences in the correction factors between these studies and ours. Nevertheless, our correction model for ESCORTAIR, with consideration of the RH level, was derived in a similar way to that with which we obtained the factor for SIDEPAK or PDR-1500. Thus, we consider no significant systematic errors to have been involved in the calculation process. A good linear relationship has been obtained between the PM$_{2.5}$ mass concentration of FEM and the responses of low-cost PM sensors as reported previously in other country [8,9,25,30]. We demonstrated the urban hotspot specific correction factor for a light-scattering sensor in Korean urban environment of interest to enable our findings and methodology to be extended and replicated by researchers who are interested in the utility of low-cost sensing device, such as ESCORTAIR, in South Korea.

OPCs are reportedly good at estimating mass if they have numerous bins, such as GRIMM [31]. However, estimating the mass concentration from a limited number of bins may be subject to a measurement error during the conversion process with the factory-provided internal conversion algorithm. Therefore, we used an additional correction factor for ESCORTAIR after comparison to GRIMM for proper usage under Korean circumstances.

It is well established that the response of monitors based on light scattering varies with aerosol size distribution, composition and optical properties and need a proper calibration process [13,32,33]. No single calibration model (or correction model) can enable accurate performance for all particle sources in microenvironments. This challenge applies to both research and consumer monitors. Although gravimetric measurements may be used to determine a source- or environment-specific calibration for a research study, the approach is not practical for routine monitoring in homes. A key objective of continuous monitoring—to activate controls—can be achieved if the monitor reliably and clearly responds to sources that account for the majority of particles in the home even if responses are not quantitative.

We conducted this study by assuming that indoor PM$_{2.5}$ emission sources exist, that is, from frying pork in a pan and smoking, with consideration of Korean life style [20] and relatively high

smoking prevalence [34]. We determined that our linear model obtained from the indoor test for each single-aerosol type showed excellent performance (R2 = 0.98 or 0.92 for ESCORTAIR; 0.94 or 0.90 for PA), compared to FEM responses. However, the responses of ESCORTAIR as well as PA were relatively less precise but good (R2 = 0.81, 0.87) for monitoring in urban traffic hotspots suggesting that IRMs need a site-specific calibration with a reference method before they are used [13,32,33].

The shape of the response curve can be related to the type of OEM sensors integrated with the monitor and relative humidity. Because we did not have information about the internal conversion factors (count to mass concentration for OPC type or light intensity to mass concentration for photometer), using GRIMM or BAM, known as US FEM, we tried to obtain our own correction factors for usage of ESCORTAIR or PA in urban indoor or outdoor settings.

The limitations of our study should be noted. First, the sample size in this study was relatively small. However, in our tests, we determined a $PM_{2.5}$ range of 10 to 3000 $\mu g/m^3$ including both indoor and outdoor measurement. This range ensures that the concentration distributions would not be systematically biased. Our $PM_{2.5}$ concentrations might not be representative of each sampling season or area as a result of spatial-temporal variations. Additional experiments are needed to understand the stability of our correction factor in different seasons and/or in other locations, that is, in industrial or rural areas of Korea. Extending the sampling periods for each season and location would ensure that our results are more representative. Furthermore, in future studies, measurements of the wind direction and speed are expected to provide improved correction factors between the IRM and FEM methods. As we mentioned in the method section, we checked flow rate for our RGMs or FEMs prior to our experiment. For IRMs, we have considered measuring flow rate but due to its open wide inlet and very low flow rate, we could not connect it to our mass flowmeter properly. Instead, especially, before our outdoor test, we operated 5 ESCORTAIRs and 5 PAs simultaneously and checked measurement errors between devices. Then, we selected 2 of them which provide best outcomes, compared to FEM. A preparation of QC/QA test program for massive products of IRMs are recommended. And for ESCORTAIR, like PA, application of weather proof design is suggested. In addition, future studies may be necessarily conducted to obtain site-specific correction factors including at coal power plants or in rural areas.

In this study, the performances of the IRM with RGM and that of the FEM operated with a high QCQA program were compared with one-hour monitoring intervals at national $PM_{2.5}$ monitoring site. This makes this study unique compared to previous studies, which were mostly conducted with one-day interval gravimetric methods at ordinary sampling sites.

Despite the growing public interest in reducing personal exposure levels to $PM_{2.5}$ in Korea, IRM monitoring still faces challenges in terms of providing real-time concentration information. Although the number of national $PM_{2.5}$ monitoring sites in Korea is increasing, additional IRMs in hotspots or communities are required because they can detect continuous spatial—temporal variations and identify nearby exposure sources on a real-time basis in a micro-environment of hotspots.

This study found that the measurement of $PM_{2.5}$ concentrations with recently developed laser-based IRM under- or over-estimates $PM_{2.5}$ concentrations obtained from FEM while its bias could be approximately 11 to 16% even at urban outdoor hotspots with traffic sources with high relative humidity levels. Therefore, the application of a correction factor is strongly suggested for inexpensive laser-based monitoring devices.

## 5. Conclusions

Our study determined that on days with a high level of outdoor RH (80% or higher), our IRMs overestimated the outdoor $PM_{2.5}$ level and showed the necessity of a correction factor for IRMs to adjust for the effect of temperature and relative humidity. $PM_{2.5}$ concentrations measured with IRMs need to be subjected to quality control and quality assurance evaluation before these monitors are used for the quantification of $PM_{2.5}$ levels in urban indoor or outdoor atmospheric environments in Korea. In consideration of relatively stable outcomes with the application of correction factors for recently

developed new IRMs—ESCORTAIR or PA—our study supports their usage in networking monitoring for various urban hotspots.

**Supplementary Materials:** The following are available online at http://www.mdpi.com/2076-3417/9/9/1947/s1. Figure S1. Scatter plots of PM$_{2.5}$ concentrations of IRMs with those of FEM, i.e., BAM, by the level of relative humidity. The data points refer to each paired hourly mean value (n = 240, except ESCORT which had 120 points due to malfunction of a device with records of high internal temperature values during our outdoor test).

**Author Contributions:** S.K. designed this study, wrote manuscript and conducted interpretation of the quantitative aspects of data analysis. S.P. performed modeling simulation and J.L. provided editorial efforts. S.K. supervised the whole study.

**Funding:** This research was funded by Environmental Health Research Center Project (2016001360002) by Korea Environmental Industry & Technology Institute, Ministry of Environment, South Korea

**Acknowledgments:** The authors deeply appreciate the technical comments for sensor evaluation from. Andrea Clements and Timothy Buckley at the national exposure research laboratory, U.S. EPA. The authors also appreciate the assistance of Sungmin Jung and the staff (Minhee Lee, Taekyung Hwang, Jieun Jeong) at the Korea PM2.5 supersite of Daejeon, South Korea for their support with data collection and the help of Juhee Kim in data screening. The content is solely the responsibility of the authors and does not necessarily represent the official views of the US EPA or Korea NIER (National Institute for Environmental Research).

**Conflicts of Interest:** The authors declare no conflict of interest.

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
