# Peer review of "Evaluation of Performance of Inexpensive Laser Based PM2.5 Sensor Monitors for Typical Indoor and Outdoor Hotspots of South Korea"

_applsci, doi:10.3390/app9091947_

Round 1

Reviewer 1 Report

The study is rather specific in terms of the indoor/outdoor scenarios chosen to addreess the measurement campaings and also limited with regard to the range of atmospheric conditions        (T, RH) considered and volume of collected data. The authors are, however, aware of this and

roperly point out and explain it in the "Discussion".

Anyway, the work is quite valuable as it contributes to the database needed to validate the performance and accuracy of IRMs against higher performance, more robust, more accurate instruments used for the same purpose.

Therefore, I consider that it is worhty to publish this work in the journal Applied Sciences due to its relevance for the gowing scientific, technical and business community involved in the development, and deployment of low-cost sensors/monitors for air quality applications.

Text checker/corrector must be applied to correct minor grammar/spelling errors.

Author Response

Thank you. As suggested we double checked the document and corrected grammar/spelling errors

Reviewer 2 Report

The current manuscript is dealing with the evaluation of performance of low cost sensors, a very interesting topic which has gained significant importance during the last yeas. A number of low cost sensors have been developed nowadays due to the needs of establishing air pollution monitoring systems as well as to perform epidemiological studies. However these kind of sensors still face serious problems with their reliability due to the very high uncertainty in their measurements. For that purpose any work trying the improve the precision, accuracy and bias of real-time low cost monitors is seriously taken under consideration. Authors  appropriate design their research methodology and the results obtained could be useful for the scientific community. However there are few points that should be addressed:

Introduction should be enriched with more references dealing with this issue. E.g Franken R. et.al 2019, Indoor Air etc.)

Better justification on the selection of sensors which were tested as well as of the reference sensors

Is the methodology used based on a description of an equivalence method and which were the deviations from this?

Authors developed their own correction models. Whats the application range and the durability, stability of the correction factors? I mean how often do you have to re-calculate the correction factors? What about the application of machine learning techniques on the correction of low cost sensors?

Author Response

Reviewer 2.

The current manuscript is dealing with the evaluation of performance of low cost sensors, a very interesting topic which has gained significant importance during the last yeas. A number of low cost sensors have been developed nowadays due to the needs of establishing air pollution monitoring systems as well as to perform epidemiological studies. However these kind of sensors still face serious problems with their reliability due to the very high uncertainty in their measurements. For that purpose any work trying the improve the precision, accuracy and bias of real-time low cost monitors is seriously taken under consideration. Authors  appropriate design their research methodology and the results obtained could be useful for the scientific community. However there are few points that should be addressed:

Q1. Introduction should be enriched with more references dealing with this issue. E.g Franken R. et.al 2019, Indoor Air etc.)

Thank you for this valuable information. We included the article written by Franken et al. in the Introduction.

L 67, A recent article reported that the one size fits all” approach to obtain PM2.5 mass concentrations by OPC result in relatively high uncertainty in complex exposure situations. Therefore, the corresponding conversion curve approach may be most valuable when a relatively high contrast is expected in exposure levels e.g. daytime home with indoor combustion sources, barbeques, or secondhand smoke versus night time, or day time outside with heavy traffic volume versus night time 16

Reference 16 :

Franken et al., Comparison of methods for converting Dylos particle number concentrations to PM2.5 mass concentrations, Indoor Air 2019;1–10.

Q2. Better justification on the selection of sensors which were tested as well as of the reference sensors

Thank you. To add to the justification, we revised objects of our study as seen below.

(After) In this study, we evaluated the performance of inexpensive (less than $300) real-time PM monitors, IRMs, with high cost (about $2,000 ~ $10,000) and cross-comparisons between them and research grade PM monitors (RGM). We used US FEMs as reference instruments (approximately $20,000 or higher) and provided a final error of mass concentration (PM2.5) measurement after applying correction factors in this study.

(Before) In this study, we conducted cross-comparisons between inexpensive (less than $300) real-time PM monitors, IRMs, with high cost (about $2,000 ~ $10,000) research grade PM monitors (RGM), and higher-cost (approximately $20,000 or higher) devices, i.e., US FEM reference instruments, in terms of performance evaluation and provided a final bias of mass concentration (PM2.5) measurement after applying the correction factor.

We have provided details of devices regarding selection of our sensors in the Methods section of our original manuscript as follows.   

2. Materials and Methods

2.1 PM2.5  real-time monitors

A laser-based light-scattering PM2.5 sensor monitor (ESCORTAIR, ESCORT, Seoul, Korea) (weight <300 g, volume <510 cm3) consisted of an optical particle counting (OPC) PM sensor(INNOSIPLE1), CO2 sensor, temperature relative humidity sensor, data transfer networking module, and light-emitting diode (LED) display screen. In the ESCORTAIR, the sensing volume is illuminated with a laser and airborne particles are counted and processed one at a time. There were various IRMs commercially available in South Korea. In this study, however, we chose ESCORTAIR as they allowed us to directly transfer data to our data server using its Wi-Fi function.

For comparison purposes, another inexpensive photometer type of PM monitor (PA, PurpleAir, CA) (https://www.purpleair.com/), mounting two Plantower sensors in a monitor, was used. PA is recommended its usage by AQ-SPEC (Air quality sensor performance evaluation center, South Coast Air Quality Management District, CA, USA) or US EPA (Environmental Protection Agency, NC, USA) as an IRM. The inlet system of these IRMs did not have an impactor or a cyclone unlike that of the RGM.

The performance of the two IRMs (one OPC, i.e., ESCORTAIR, and one photometer, i.e., PA) costed less than $300 were simultaneously compared with those of high-cost devices ($10,000 or so), i.e., research-grade laser photometers including PDR-1500 (Thermo Scientific, MA) and SIDEPAK AM510 (TSI, Inc., Shoreview, MN, USA).

We used the U.S. EPA Federal Equivalent Method (FEM), i.e., GRIMM180 (GRIMM, Germany) for indoor test and BAM-1022 (MetOne, OR) for outdoor field tests16.

Grimm Technologies, Inc. Model EDM 180 PM2.5 Monitor is light scattering OPC monitor operated for 24 hours at a volumetric flow rate of 1.2 L/min, configured with a Nafion®- type air sample dryer. BAM-1022, a beta-ray attenuation mass monitor has a PM2.5 particle size separator. Using BAM, we obtained 24 1-hour average measurements at the national PM2.5 monitoring supersite operated by National Institute of Environmental Research, (Daejeon, Korea).

Q3. Is the methodology used based on a description of an equivalence method and which were the deviations from this?

We appreciate this question. As previously mentioned in Table 1 of our original manuscript and method section, we had two typical PM2.5 laser sensor types of IRMs; ESCORT, OPC type, and PA, photometer type.

Since US EPA announced measuring PM2.5 with either GRIMM EDM180 (OPC type) or BAM (photometer type employing the absorption of beta radiation) as a federal equivalent method (https://www3.epa.gov/ttnamti1/files/ambient/criteria/AMTIC%20List%20Dec%202016-2.pdf), we used them as our standard methods.

Because the dimensions (h x w x d) of GRIMM EDM180, operated with OPC type sensor, was 26.6 x 48.3 x 36.4 cm and was semi portable, we could install it in our own indoor laboratory.

However, BAM, another federal equivalent method operated with beta-ray, is recommended by manufacturers as bench-top or rack mount, usually installed inside a “walk-in shelter”. We could not install it in our laboratory due to our research budget and time. For this study, instead of installling our own BAM facility in our lab, we used outcomes of a BAM installed in a national PM2.5 monitoring shelter in South Korea where we measured outdoor PM2.5 on the roof of the shelter with IRMs.

We believe that our methodology and/or study design can be an example of this type of study evaluating performance of sensor based monitors. 

Finally, as result of our test, we also have provided difference of IRMs compared to FEM in our original manuscript as seen below.

L263. Between the two IRMs and FEM, the Spearman correlation coefficients were 0.97 (P = 0.0001) or 0.92 (P = 0.0001) between ESCORTAIR and GRIMM for indoor pan-frying or the SHS test, respectively. The correlation coefficients between PA and GRIMM were similar (0.97 (P = 0.0001) or 0.86 (P = 0.0001)) for the indoor test (Table 2). Our outdoor test also showed high correlation between ESCORTAIR and BAM (0.84 (P = 0.0001)). The correlation coefficient for PA was 0.88 (P = 0.0001).

L317. We found the difference (median (IQR)) with the calibrated data, compared to FEM, to be 16.3. (8.5 ~ 28.0) % for ESCORTAIR and 14.5 (6.1 to 23.5) % for PA for outdoor environments (Figure 3).

Q4. Authors developed their own correction models. Whats the application range and the durability, stability of the correction factors? I mean how often do you have to re-calculate the correction factors? What about the application of machine learning techniques on the correction of low cost sensors?

Thank you for your valuable comment. We developed our model using data collected over Spring and Summer to cover our rainy season. For indoor data, we believe we can apply our results for four season since indoor temperature and relative humidity levels are relatively stable. In the case of outdoor results, we may need to recalculate value for Winter . We already mentioned this in the Discussions section. In studies in the near future, we will develop and apply machine learning techniques on the correction of low cost sensors.

L407. Additional experiments are needed to understand the stability of our correction factor in different seasons and/or in other locations, i.e., in industrial or rural areas of Korea.